# Chromatin interaction maps identify Wnt responsive *cis*-regulatory elements coordinating *Paupar-Pax6* expression in neuronal cells

**Ioanna Pavlaki**[1,☯], **Michael Shapiro**[1,2,☯], **Giuseppina Pisignano**[1], **Stephanie M. E. Jones**[1], **Jelena Telenius**[3], **Silvia Muñoz-Descalzo**[1,4], **Robert J. Williams**[1], **Jim R. Hughes**[3], **Keith W. Vance**[1]*

1 Department of Biology and Biochemistry, University of Bath, Bath, United Kingdom, 2 The Francis Crick Institute, London, United Kingdom, 3 MRC Molecular Haematology Unit, Weatherall Institute of Molecular Medicine, Radcliffe Department of Medicine, University of Oxford, Oxford, United Kingdom, 4 Instituto Universitario de Investigaciones Biomédicas y Sanitarias, Universidad Las Palmas de Gran Canaria, Las Palmas de Gran Canaria, Spain

☯ These authors contributed equally to this work.
* k.w.vance@bath.ac.uk

**Data Availability Statement:** All multiplexed NG Capture-C DNA sequencing data for this project have been deposited in the NCBI GEO database

## Abstract

Central nervous system-expressed long non-coding RNAs (lncRNAs) are often located in the genome close to protein coding genes involved in transcriptional control. Such lncRNA-protein coding gene pairs are frequently temporally and spatially co-expressed in the nervous system and are predicted to act together to regulate neuronal development and function. Although some of these lncRNAs also bind and modulate the activity of the encoded transcription factors, the regulatory mechanisms controlling co-expression of neighbouring lncRNA-protein coding genes remain unclear. Here, we used high resolution NG Capture-C to map the *cis*-regulatory interaction landscape of the key neuro-developmental *Paupar-Pax6* lncRNA-mRNA locus. The results define chromatin architecture changes associated with high *Paupar-Pax6* expression in neurons and identify both promoter selective as well as shared *cis*-regulatory-promoter interactions involved in regulating *Paupar-Pax6* co-expression. We discovered that the TCF7L2 transcription factor, a regulator of chromatin architecture and major effector of the Wnt signalling pathway, binds to a subset of these candidate *cis*-regulatory elements to coordinate *Paupar* and *Pax6* co-expression. We describe distinct roles for *Paupar* in *Pax6* expression control and show that the *Paupar* DNA locus contains a TCF7L2 bound transcriptional silencer whilst the *Paupar* transcript can act as an activator of *Pax6*. Our work provides important insights into the chromatin interactions, signalling pathways and transcription factors controlling co-expression of adjacent lncRNAs and protein coding genes in the brain.

(https://www.ncbi.nlm.nih.gov/geo/) under accession number GSE129697. The data underlying the RT-qPCR and luciferase assay results are provided as a Supporting Information file (S9 Table).

**Funding:** This work was supported by a Biotechnology and Biological Sciences Research Council (BB/N005856/1; IP, MS, KWV) grant awarded to KWV. The funder had no role in study design, data collection and analysis, decision to publish, or preparation of the manuscript. BBSRC website: https://bbsrc.ukri.org/.

**Competing interests:** The authors have declared that no competing interests exist.

## Author summary

Long non-coding RNA (lncRNA) genes are often co-expressed in the brain with neighbouring protein coding genes involved in gene expression control. Such lncRNA-protein coding gene pairs are predicted to work together to regulate neuronal development and function. Despite this, the regulatory mechanisms controlling their co-expression is poorly understood. Here, we identify the chromatin interactions and DNA sequences controlling expression of the *Paupar-Pax6* lncRNA-protein coding locus in the brain. We show that the Wnt signalling pathway, a key regulator of development and disease, acts through the TCF7L2 transcription factor to co-ordinate their expression. We find that *Paupar* DNA contains sequences that silence *Pax6* whilst the *Paupar* RNA can act as an activator of *Pax6*. Our work generates new insights into the complex regulatory relationship controlling the function of neighbouring lncRNA-protein coding genes and is important for understanding development of the brain.

## Introduction

A typical gene promoter is regulated by multiple different types of *cis*-regulatory elements (CREs) such as transcriptional enhancers and silencers. These are DNA sequences containing clusters of transcription factor binding sites that act together to generate the correct temporal and spatial expression of their target genes [1]. Chromatin conformation capture (3C) based technologies have shown that short- and long-range dynamic chromatin looping interactions bring CREs and their target promoters into close physical proximity in the nucleus to facilitate gene regulation. More recently, high throughput 3C variants such as NG Capture-C have been used to map large numbers of CREs to their cognate genes at high resolution and investigate the complexity of CRE-promoter communication at unprecedented detail [2,3].

Precise temporal and spatial control of expression of the *Pax6* transcription factor gene is required for the normal development and function of the nervous system. *Pax6* haploinsufficiency in mice results in abnormal eye and nasal development and causes a range of brain defects; whilst mutations affecting *PAX6* expression and function in humans cause anirida, an autosomal dominant inherited disorder characterized by a complete or partial absence of the iris [4–6]. *Pax6* is transcribed from 2 major upstream promoters (P0, P1) and multiple CREs have been shown to control *Pax6* expression in distinct domains in the central nervous system and eye [7,8]. These include the neuroretina, ectodermal and retinal progenitor enhancers just upstream of the P0 promoter [9,10]; a photoreceptor enhancer situated between the P0 and P1 promoters [8] the retina regulatory region located between exons 4 and 5 [8,9]; and three conserved sequence elements within intron 7 that activate *Pax6* in the diencephalon, rhombencephalon and at late stages of eye development [11]. These CREs are all located within a 30 kb window surrounding the *Pax6* P0 and P1 promoters and act over short genomic distances. In addition, several candidate long-range enhancers have been identified approximately 150–200 kb downstream of the *Pax6* gene and some of these have also been shown to drive *Pax6* expression in specific domains of the eye and brain [12,13]. However, these enhancers together are not sufficient to generate the full temporal and spatial pattern of *Pax6* expression in the central nervous system suggesting the presence of additional uncharacterised *Pax6* regulatory elements.

Thousands of long non-coding RNAs (lncRNAs) are temporally and spatially expressed within the central nervous system and some of these are thought to be important in brain development and function [14–16]. Brain-expressed lncRNAs are preferentially located in the

genome close to protein coding genes involved in transcriptional control [17]. This includes bidirectional lncRNAs that are transcribed in the opposite direction to a protein coding gene from a shared promoter as well as intergenic lncRNAs that are either expressed from their own promoter or from a transcriptional enhancer. Such lncRNA-mRNA pairs are frequently co-expressed during neuronal development and in different brain regions and can function in the control of similar biological processes [18,19]. The lncRNA *Paupar*, transcribed from a promoter approximately 8.5 kb upstream of the *Pax6* gene, is an important regulator of neurogenesis *in vivo* in mouse and human, and is co-ordinately expressed with *Pax6* during neural differentiation *in vitro* and in the adult mouse brain [19,20]. Moreover, *Paupar* transcript directly binds PAX6 and acts as a transcriptional cofactor to promote the formation of a PAX6-*Paupar*-KAP1 chromatin regulatory complex at important neuronal genes [19,20]. Even though *Paupar* and *Pax6* can act together to regulate shared biological processes important for neuronal development, the CREs controlling *Paupar-Pax6* co-expression in the nervous system are not known.

High resolution NG Capture-C to generate high resolution chromatin interaction maps with the *Paupar* and *Pax6* promoters in *Paupar-Pax6* high- and low-expressing cells. The results identified shared chromatin interactions with both the *Paupar* and *Pax6* promoters involved in regulating *Paupar-Pax6* co-expression, as well as promoter specific *cis*-regulatory interactions. We discovered transcription factor motifs within a prioritised set of chromatin interactions and show that the Wnt signalling pathway acts through TCF7L2 to co-ordinate *Paupar* and *Pax6* co-expression in neuronal cells. The results demonstrate that the *Paupar* DNA locus contains a TCF7L2 bound transcriptional silencer whilst the *Paupar* transcript can activate *Pax6* expression, defining distinct roles for *Paupar* in *Pax6* expression control. We also report cell type specific differences in both local and distal chromatin interactions with the *Paupar* and *Pax6* promoters that may be important for CRE-promoter communication and *Paupar-Pax6* activation in neurons. Our work further refines the complex *cis*-regulatory landscape surrounding the *Paupar-Pax6* locus and provides critical insights into the regulatory mechanisms controlling the co-expression of adjacent lncRNAs and protein coding genes in the brain.

## Results

### Identification of *cis*-regulatory interactions with the *Paupar* and *Pax6* promoters using high resolution NG Capture-C

Expression of the lncRNA *Paupar* and its adjacent *Pax6* transcription factor gene are temporally coordinated during *in vitro* neural differentiation of mouse embryonic stem cells (ESCs); whilst *Paupar* and *Pax6* are mostly highly expressed in the adult brain compared to other mouse tissues [19]. Despite this, the chromatin interactions, signalling pathways and transcription factors controlling *Paupar-Pax6* co-expression in the brain remain poorly understood. We therefore investigated *Paupar-Pax6* expression control in the neuronal lineage using the following cell types: primary neural stem cells (NSCs) isolated from E14.5 mice, differentiated mouse cortical neurons, N2A mouse neuroblastoma cells, as well as mouse ESCs as a non-neuronal reference. RT-qPCR analysis demonstrated that *Paupar* and *Pax6* P0 and P1 expression is significantly higher in neuronal cell types compared to ESCs, with highest expression in NSCs and differentiated neurons (Fig 1A and 1B). Consistent with an earlier report [21], our results also suggest that the *Pax6* P1 promoter is the major *Pax6* promoter in the neuronal lineage. We found that *Pax6* P1 promoter transcription was 40- and 105-fold more active than P0 in NSCs and differentiated neurons respectively, whilst in N2A cells *Pax6* P0 expression was undetectable (Fig 1A and 1B).

High resolution NG Capture-C was performed to map chromatin interactions with the *Paupar* and *Pax6* P0 and P1 promoters, as well as the *Sox2* promoter as a positive control, to

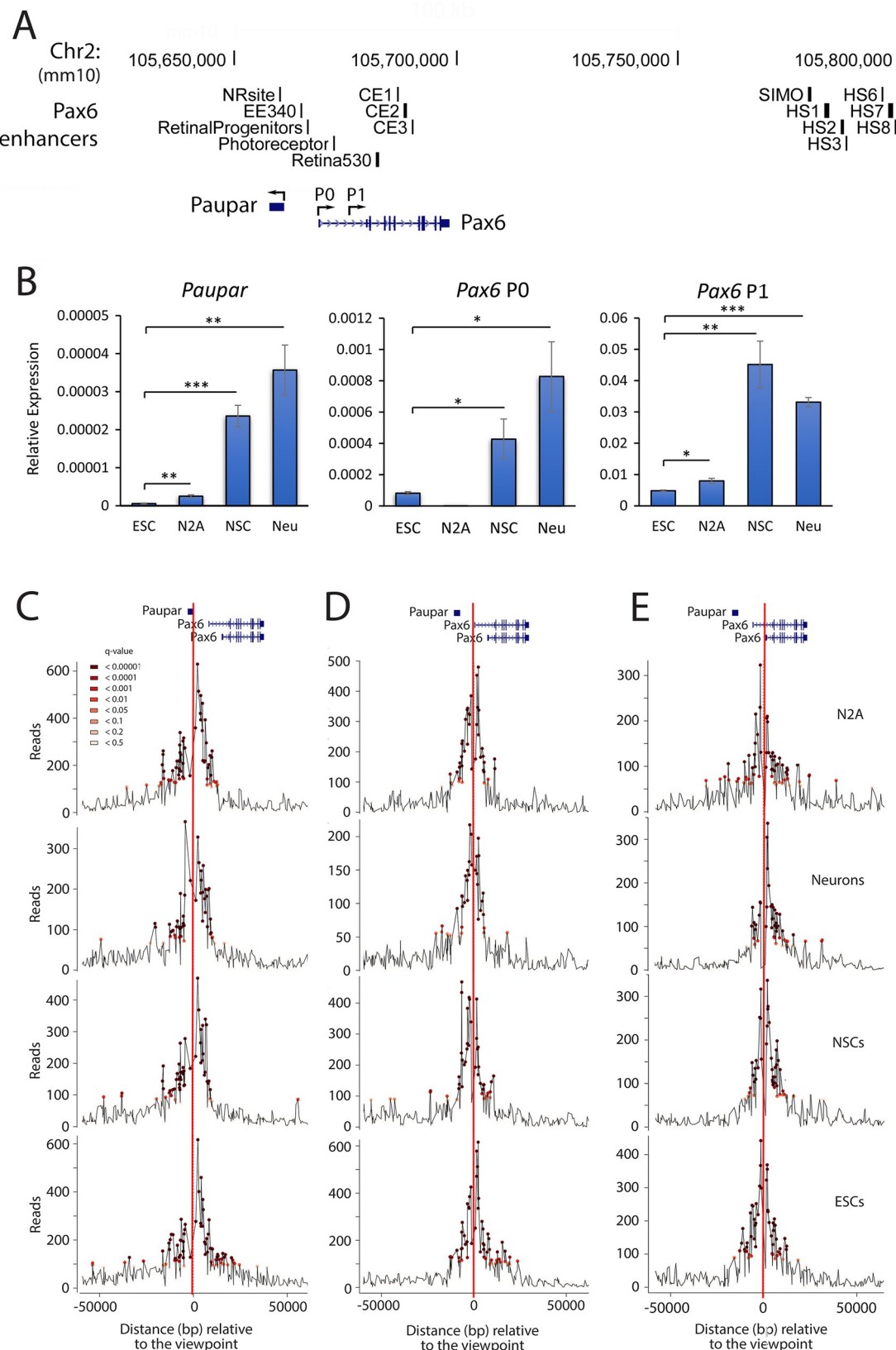

**Fig 1. High resolution *cis*-regulatory interactions with the *Paupar* and *Pax6* promoters in *Paupar-Pax6* high- and low-expressing cells.** (A) Genome browser graphic (GRCm38/mm10) showing the location of *Paupar* and *Pax6* genes, the two major *Pax6* promoters (P0 and P1) and known *Pax6* enhancers. NR (neuroretina), EE (ectodermal enhancer), RP (retinal progenitors), PR (photoreceptor), Re (retina), CE1-3 (conserved element 1–3), HS (hypersensitivity site). (B) Transcripts generated from the *Paupar* and *Pax6* P0 and P1 promoters were measured in ESCs, NSCs, differentiated neurons (Neu) and N2A cells using RT-qPCR. Results are presented relative to the *Tbp* reference gene. Mean values +/- sem shown, n = 6, one-tailed t-test, unequal variance. * p<0.05, ** p<0.01, *** p<0.001. (C, D, E) NG Capture-C profiles displaying the interaction count per DpnII restriction enzyme fragment for each NG Capture-C library. The red vertical lines indicate the locations of captured viewpoints. (C) *Paupar* promoter, (D) *Pax6* P0 and (E) *Pax6* P1 promoter viewpoints. Significant fragments determined using r3C-seq [22] are denoted by coloured circles.

identify the *cis*-regulatory DNA sequences important for *Paupar-Pax6* expression control and the chromatin changes associated with activation of the locus in neuronal cells. Multiplexed NG Capture-C libraries were generated and sequenced to an average depth of 61 million paired end reads per library. Benchmarking NG Capture-C data quality using CCanalyser [3] showed that an average of 37.6% mapped reads contained capture bait sequence across all NG Capture-C libraries, demonstrating good capture enrichment, and confirmed good ligation efficiency as 29.3% of captured fragments were ligated to a reporter (S1 Table). This enabled us to generate high resolution NG Capture-C interaction profiles (Figs 1C, 1D and 1E and S1) using an average of 33,719 (S2 Table) unique interactions between DpnII restriction fragments and each promoter bait fragment per cell type.

NG Capture-C interaction profiles were then analysed using r3C-seq tools [22] to normalise for distance from the capture point and model statistically significant (q < 0.1) CRE-promoter interactions. As expected, we identified significant looping interactions between the *Sox2* super-enhancer overlapping the *Peril* locus [23] and the *Sox2* promoter in ESCs which were not present in neuronal cell types (S1 Fig). This is consistent with previous 3C based maps defining *Sox2* enhancer-promoter communication [23] and confirms the ability of our approach to identify functional CREs. To define the set of regulatory interactions mediating *Paupar-Pax6* expression we then determined the number of statistically significant chromatin interactions with the *Paupar* and *Pax6* P0 and P1 promoters present in both biological replicates for each cell type. We discovered that 96% high resolution of chromatin interactions are located within a 50 kb window centred around each promoter (Fig 2A), including both upstream and downstream *cis*-acting DNA sequences. The *Paupar-Pax6 cis*-regulatory interaction map showed significant interactions with known *Pax6* regulatory elements as well as many additional short-range regulatory interactions with candidate new CREs involved in *Paupar-Pax6* expression control (Fig 2B and S3 Table for fragment coordinates). Consistent with a role in neuronal gene expression, chromatin interactions with the *Paupar* and *Pax6* promoters show an increased association with H3K4me1 ChIP-seq peaks in E12.5 mouse forebrain tissue compared to ESCs (Fig 2B) using publicly available ENCODE data [24]. Moreover, permutation testing revealed statistically significantly enrichment (p < 0.05) in the overlap between forebrain H3K4me1 peaks and chromatin interactions with the *Pax6* promoter viewpoints in all four cell types (Fig 2B and S4 Table). No significant intersection (p<0.05) was detected between ESC H3K4me1 peaks and *cis*-interactions with any of the promoter viewpoints (Fig 2B). As H3K4me1 marks both active and poised enhancers this implies that the NG Capture-C interactions may function to control expression of the *Paupar-Pax6* locus in neuronal cells [25]. The interaction map also encompasses associations between DNA sequence elements within the *Paupar* genomic locus and the *Pax6* promoters (Fig 2B and S5 Table). Most chromatin interactions (119 out of 168) were found in all cell types tested but we also discovered a subset of neuronal (30/168) and ESC (19/168) specific interactions that may be important for tissue specific regulation of the locus (S6 Table). Our results revealed shared

A

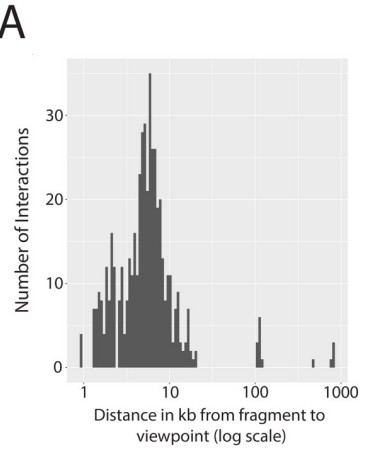

B

**REPRODUCIBLE *CIS*-REGULATORY INTERACTIONS WITH EACH PROMOTER**

|  | ESC | NSC | Neuron | N2A |
|---|---|---|---|---|
| *Paupar* TSS | 49 | 27 | 40 | 44 |
| *Pax6* P0 | 33 | 30 | 28 | 30 |
| *Pax6* P1 | 42 | 32 | 34 | 27 |

***CIS*-INTERACTIONS OVERLAPPING H3K4ME1 CHROMATIN IN ESCs**

|  | ESC | NSC | Neuron | N2A |
|---|---|---|---|---|
| *Paupar* TSS | 4 | 1 | 4 | 5 |
| *Pax6* P0 | 6 | 5 | 6 | 5 |
| *Pax6* P1 | 8 | 8 | 8 | 5 |

***CIS*-INTERACTIONS CORRESPONDING TO KNOWN *PAX6* LOCAL ENHANCERS**

|  | ESC | NSC | Neuron | N2A |
|---|---|---|---|---|
| *Paupar* TSS | 5 | 4 | 5 | 5 |
| *Pax6* P0 | 6 | 8 | 6 | 6 |
| *Pax6* P1 | 6 | 5 | 4 | 5 |

***CIS*-INTERACTIONS BETWEEN PAUPAR LOCUS AND INDICATED PROMOTERS**

|  | ESC | NSC | Neuron | N2A |
|---|---|---|---|---|
| Paupar TSS | 2 | 2 | 2 | 2 |
| Pax6 P0 | 0 | 3 | 1 | 1 |
| Pax6 P1 | 1 | 0 | 0 | 0 |

***CIS*-INTERACTIONS OVERLAPPING H3K4ME1 CHROMATIN IN THE BRAIN**

|  | ESC | NSC | Neuron | N2A |
|---|---|---|---|---|
| *Paupar* TSS | 22 | 10 | 14 | 16 |
| *Pax6* P0 | 23* | 21* | 18* | 20* |
| *Pax6* P1 | 34* | 26* | 29* | 20* |

**REPRODUCIBLE *TRANS* INTERACTIONS WITH DNA SEQUENCES ON OTHER CHROMOSOMES**

|  | ESC | NSC | Neuron | N2A |
|---|---|---|---|---|
| *Paupar* TSS | 4 | 5 | 5 | 0 |
| *Pax6* P0 | 5 | 4 | 4 | 0 |
| *Pax6* P1 | 5 | 6 | 7 | 0 |

C

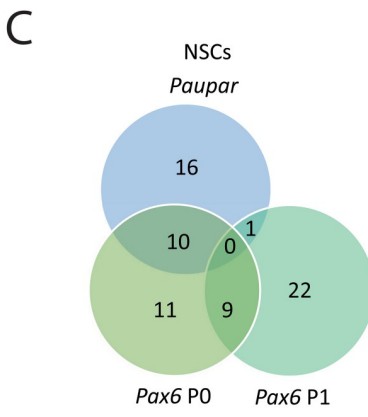

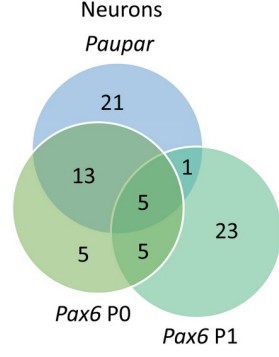

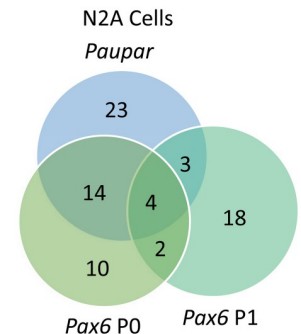

D

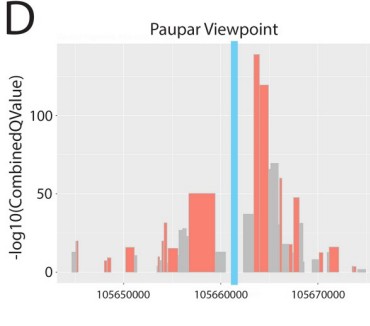

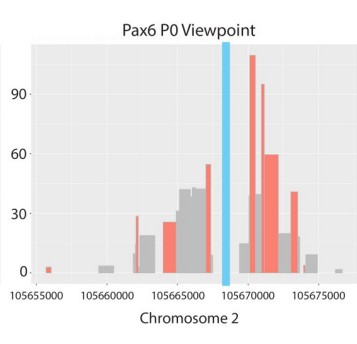

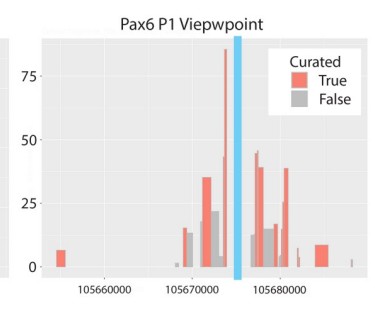

E

**Fig 2. Identification of short-range regulatory interactions with candidate new CREs involved in *Paupar-Pax6* expression control.** (A) Histogram depicting distances of *cis*-interacting fragments from their viewpoints. Significant duplicated fragments from all cell types and viewpoints are shown. (B) Tables showing the number of statistically significant reproducible interactions with the indicated promoter viewpoints as well as overlap with known *Pax6* enhancers and enhancer-like chromatin marks using ENCODE data [24]. Permutation testing to test for significance in the overlap between chromatin interactions and H3K4me1 ChIP-seq peaks * p<0.05. (C) Venn diagram displaying the number of shared and specific interactions with the *Paupar*, *Pax6* P0 and *Pax6* P1 promoters in NSCs (left), neurons (middle) and N2A cells (right). (D) Curation of interacting fragments for the indicated viewpoints based on combined–log 10 q value from NG Capture-C N2A cell data. q values from replicates were merged by taking the square root of their product. Each bar represents a fragment with significant interaction (q < 0.05). (E) 3D model of *Paupar-Pax6* local chromatin architecture generated from NG Capture-C N2A cell data using the 4Cin software package [26]. Viewpoints and curated fragments are marked using the designated colours. The model is shown from three angles.

interactions with more than one promoter region that may be important for *Paupar-Pax6* co-expression in the brain (Fig 2C). In addition, we found a subset of specific *cis*-regulatory interactions with individual *Paupar*, *Pax6 P0* and *P1* promoter viewpoints, suggesting that *Paupar* and *Pax6* expression control may be decoupled (Fig 2C), as well as a small number of *trans*-interactions with DNA sequences on different chromosomes (Fig 2B and S3 Table). Altogether, these results identify the chromatin interactions and CREs that are likely to be important for precise *Paupar-Pax6* expression control.

## Wnt signalling acts through TCF7L2 to regulate co-expression of the *Paupar-Pax6* locus

We hypothesized that a subset of discovered genomic fragments would play a causal role in the formation of chromatin interactions needed for expression of the the *Paupar-Pax6* locus and that these would have an elevated discovery rate in our analysis. N2A cells were used to identify and define the function of such sequences. These cells represent a well characterised tractable *in vitro* model of neuronal differentiation and were previously used to determine *Paupar* and *Pax6* gene regulatory functions [19,20]. To identify sequences with increased proximity to the promoter viewpoints compared to surrounding sequence, we plotted the mean -log10 q-value for each reproducible *cis*-regulatory interaction against chromosome position using N2A cell NG Capture-C data. We next determined the DpnII fragments with the highest local statistical significance and curated a subset of 42 unique fragments (24 for *Paupar*, 10 for *Pax6* P0 and 22 for the *Pax6 P1* viewpoint), visualised as peaks of increased statistical prevalence (Fig 2D). 4Cin software [26] was then applied to generate a 3D model of *Paupar-Pax6* local chromatin architecture from N2A cell NG Capture-C data and visualise the relative proximity of the curated fragments to the *Paupar* and *Pax6* promoters. The results showed that most curated fragments are located at curvature points on the chromatin fibre and appear to be orientated towards the *Paupar-Pax6* promoters (Fig 2E). This subset of curated NG Capture-C fragments may thus represent candidate *Paupar-Pax6* CREs with roles in the regulation of short-range chromatin interactions.

CRE-promoter communication is mediated by protein-protein interactions between transcription factors bound to specific motifs within CREs and proteins assembled at the target promoters. To investigate the transcription factors controlling *Paupar-Pax6* co-expression in the brain we used a custom motif discovery tool to search for transcription factor position frequency matrices (PFMs) within the N2A cell curated fragment dataset that have a high likelihood of factor binding (as described in [27]). This identified 269 high scoring transcription factor motifs that occur more than 10 times within the set of curated fragments (listed in S7 Table). We selected five of these transcription factors (KLF16, TCF7L2, ARID3A, MASH1 (ASCL1) and SOX6) with known functions in neuronal development to test for roles in *Paupar-Pax6* expression control. To do this, we transfected N2A cells with specific

endoribonuclease-prepared pools of siRNAs (esiRNAs) targeting these transcription factors and measured changes in *Paupar* and *Pax6* expression using RT-qPCR. The results showed that depletion of *Tcf7L2* and *Mash1* (*Ascl1*) led to a significant decrease in both *Paupar* and *Pax6* (Fig 3A) whilst silencing *Klf16*, *Sox6* and *Arid3a* increased *Paupar* but not *Pax6* expression (Fig 3B). This suggests that *Tcf7L2* and *Mash1* (*Ascl1*) are involved in coordinating *Paupar* and *Pax6* activation in neuronal cells and that *Klf16*, *Sox6* and *Arid3a* act to selectively silence *Paupar*.

We prioritised *Tcf7L2* for further investigation as it is a major effector of the Wnt/β-catenin key neuro-developmental signalling pathway and an important regulator of chromatin structure. To determine the role of *Tcf7L2* in *Paupar-Pax6* expression control we first manipulated Wnt signalling and measured changes in *Tcf7L2*, *Paupar* and *Pax6* levels using RT-qPCR in N2A cells. Activation of canonical Wnt signalling using either ectopic expression of a constitutively active β-catenin S33Y protein [28] or recombinant WNT3a ligand led to a significant reduction in *Tcf7L2* expression and a concomitant decrease in *Paupar* and *Pax6* levels after 72 hours (Fig 3C and 3D). Treatment of N2A cells with BMP4, a component of the Bmp pathway that is known to cross-talk with Wnt signalling in neuroblastoma [29], led to a 2.3-fold increase in *Tcf7l2* and a subsequent 2.1- and 1.9-fold up-regulation in *Paupar* and *Pax6* expression 72 hours later (Fig 3E). Finally, treatment of N2A cells with RSPO2, a leucine rich repeat-containing G-protein coupled receptor (LGR) ligand that has been shown to amplify Wnt signal in a subset of neuroblastoma cells [30], did not induce significant changes in either *Tcf7L2*, *Paupar* or *Pax6* (Fig 3F). These data show that changes in *Tcf7L2* levels are accompanied by reciprocal alterations in *Paupar* and *Pax6* expression and are consistent with a model in which Wnt signalling acts through TCF7L2 to co-ordinately regulate both *Paupar* and *Pax6* expression in neural cells.

## Novel TCF7L2 bound CREs control *Paupar-Pax6* co-expression

We next mapped TCF7L2 chromatin occupancy at its predicted motifs within the *Paupar-Pax6* regulatory region and annotated the function of candidate CREs encompassing these binding sites. Our motif search discovered 13 high scoring PFMs for the TCF7L2 transcription factor (S7 Table) within curated NG Capture-C fragments and qPCR primers were designed to amplify 7 candidate CREs (CRE1-7) encompassing these motifs (Fig 4A). ChIP-qPCR showed that TCF7L2 binding was 2.3-fold, 3.9-fold, 20-fold, and 11-fold enriched at CRE1, CRE2, CRE3 and CRE5 respectively, compared to an IgG isotype control (Fig 4B). TCF7L2 chromatin occupancy was not enriched at CRE4, CRE6, CRE7 and two regions that do not span any TCF7L2 motifs confirming the specificity of the experiment.

We then performed CRISPR interference (CRISPRi) to study the function of the TCF7L2 bound CREs within their endogenous chromatin context in *Paupar-Pax6* expression control in N2A cells. To do this, single guide RNAs (sgRNAs) targeting either the TCF7L2 motif in each ChIP-defined TCF7L2 bound CRE or nearby non-TCF7L2 motif containing sequences (Fig 4A), were used to recruit a catalytically inactive dCas9-KRAB fusion protein to induce local chromatin closing and block regulatory element activity [31]. RT-qPCR analysis of *Paupar* and *Pax6* expression showed that targeting dCas9-KRAB to CRE3 induced a significant 22% reduction in *Paupar* and a 31% reduction in *Pax6* levels (Fig 4C). CRISPRi against the TCF7L2 motif in CRE2 led to a 2.1-fold up-regulation of *Paupar* and a 1.4-fold increase in *Pax6* expression (Fig 4C); whilst inhibition of CRE5, located within the *Paupar* DNA locus, resulted in a 2.0-fold increase in *Paupar* and a 1.5-fold increase in *Pax6* expression (Fig 4C). dCas9-KRAB recruitment to CRE1 and two nearby regions within the *Paupar-Pax6* regulatory region that do not contain TCF7L2 motifs did not lead to coordinated changes in *Paupar-*

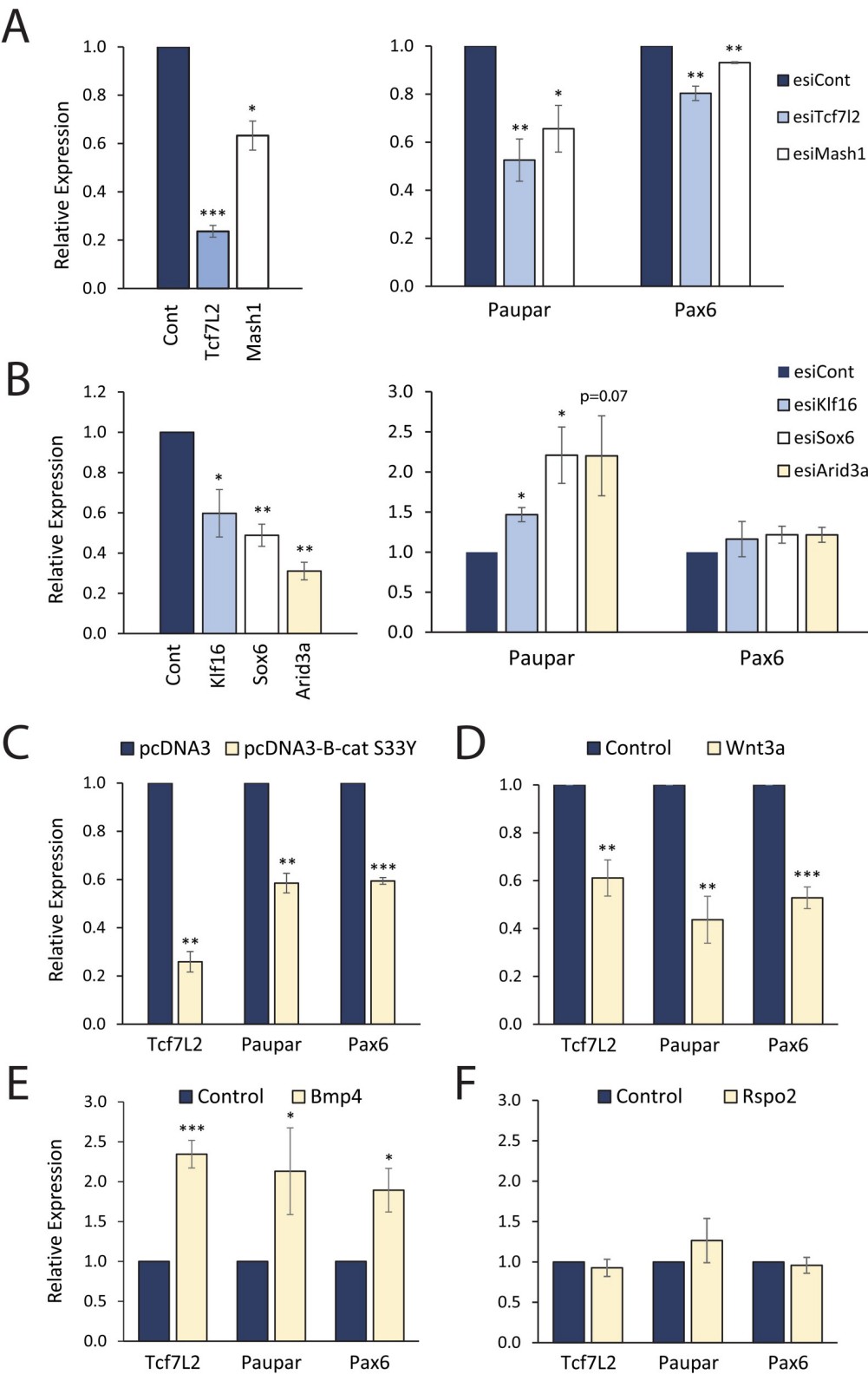

**Fig 3. Identification of transcription factors and signalling pathways controlling *Paupar-Pax6* expression in neuronal cells.** (A) *Tcf7L2*, *Mash1* and (B) *Klf16*, Sox6 and *Arid3a* expression were knocked-down in N2A cells by transfection of endoribonuclease-prepared pools of esiRNAs. An esiRNA pool targeting the luciferase gene was used as a negative control. Cells were harvested for expression analysis 3 days after transfection. (C) N2A cells were transfected

with pCl β-catenin S33Y or pCl empty vector and harvested for expression analysis 2 days later. Results are presented relative to the pCl control. (D, E, F) N2A cells were treated with either 100 ng/μl WNT3A (D), 5 ng/ml BMP4 (E) or 100 ng/μl RSPO2 (F) and harvested for RT-qPCR analysis 3 days later. 0.02% BSA in PBS was used as a negative control. For RT-qPCR reactions: target gene expression was measured using RT-qPCR and results were normalised to *Tbp*. Results are presented as mean values +/- sem, n≥3. One-tailed student's t-test * $p<0.05$, ** $p<0.01$, *** $p<0.001$.

*Pax6* expression. Taken together, these results suggest that the CRE3 TCF7L2 motif functions as part of a shared transcriptional enhancer of both *Paupar* and *Pax6*, that the CRE2 and CRE5 TCF7L2 motif containing CREs co-ordinately repress both *Paupar* and *Pax6*, and that the *Paupar* DNA locus itself plays a regulatory role in *Pax6* expression control.

### *Paupar* transcript activates *Pax6*

We previously showed that shRNA mediated depletion of *Paupar* in N2A cells increased *Pax6* levels [19]. Whilst this suggests that *Paupar* transcript silences *Pax6*, interpretation is complicated by the fact that PAX6 can bind its own promoter and negatively autoregulate its own expression in a context dependent manner [32]. Furthermore, these data are not fully consistent with the perturbation experiments here showing positive correlation between changes in *Paupar* and *Pax6*. We therefore investigated the intrinsic transcriptional regulatory function of *Paupar* using N2A cell reporter assays to gain a better understanding of *Paupar* transcript-dependent regulation of *Pax6*. To do this, MS2-tagged *Paupar* was recruited to UAS sites upstream of a *Pax6* promoter reporter using a Gal4-MS2 phage coat protein (MCP) fusion protein in a transient transfection assay (Fig 4D). This resulted in a statistically significant ($P = 0.006$) 60% increase in *Pax6* promoter activity. Removal of Gal4-MCP, so that *Paupar* was no longer recruited to the reporter, led to a significant reduction in the *Paupar* transcriptional response whilst expression of Gal4-MCP fusion protein on its own had no effect on *Pax6* promoter activity, confirming the specificity of the assay. Together with the findings in [32], these data are consistent with a model in which *Paupar* and *Pax6* expression is coordinated and that *Paupar* transcript up-regulates *Pax6* levels whilst PAX6 protein fine tunes its own expression through a negative auto-regulatory feedback loop in N2A cells.

### Identification of cell type specific chromatin architecture changes associated with high *Paupar* and *Pax6* expression in neurons

Comparative analyses of NG Capture-C profiles have previously been used to investigate *cis*-regulatory mechanisms controlling cell type specific gene expression [3]. We therefore compared NG Capture-C profiles from *Paupar-Pax6* high-expressing differentiated neurons with low-expressing ESCs to detect changes in chromatin architecture associated with increased *Paupar-Pax6* expression in the brain. To do this, normalised NG Capture-C data were first grouped into bins of discrete sizes to increase signal over noise. This showed that a 10 kb bin size facilitated the identification of changes in chromatin conformation due to clustering of neighbouring fragments that were not significant at the individual fragment level (S2 Fig). Furthermore, these changes were not discovered using permuted data validating specificity. We then examined changes in local chromatin architecture surrounding the *Paupar-Pax6* locus up to +/- 50kb from each viewpoint (Fig 5A). Permutation testing to compare the frequency of upstream versus downstream chromatin interactions identified a significant increase in upstream chromatin interactions with the *Pax6* P1 promoter in neurons compared to ESCs and an increase in downstream chromatin interactions with the *Pax6* P1 promoter in neurons compared to ESCs (Fig 5A). On the other hand, genomic sequences upstream of the Pax6 P1 promoter show significantly increased chromatin interactions with the *Paupar* viewpoint in

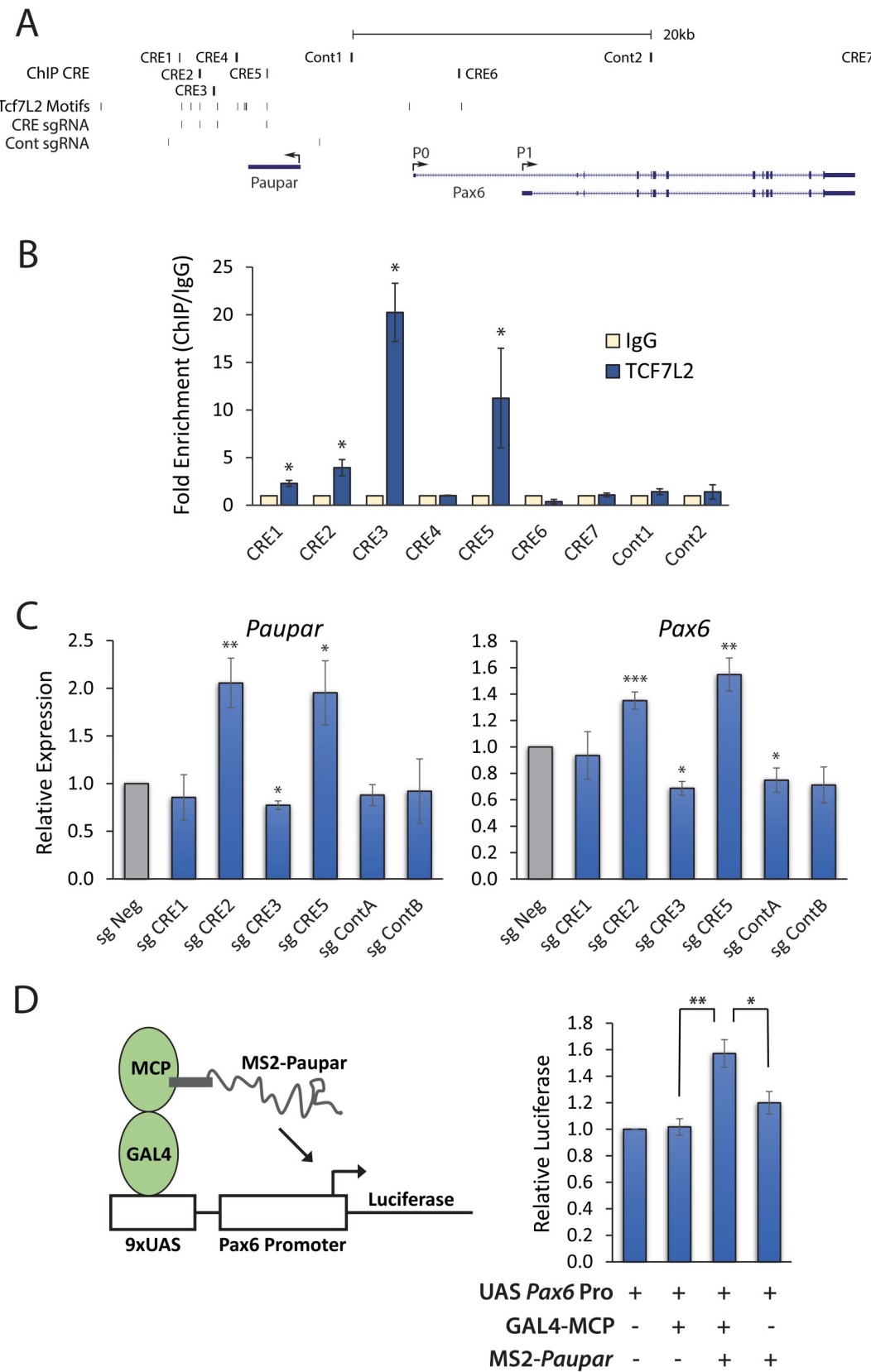

**Fig 4. TCF7L2 bound CREs coordinate *Paupar-Pax6* expression in neuronal cells.** (A) Genome browser graphic showing the ChIP amplified candidate CREs as well as the location of the TCF7L2 motifs and sgRNAs used in the CRISPRi experiment (GRCm38/mm10). (B) ChIP assays were performed in N2A cells using either an antibody against TCF7L2 or an isotype specific control. TCF7L2 occupancy at the indicated CREs was analysed by qPCR. Fold enrichment was calculated as $2^{-\Delta\Delta Ct}$ (IP/IgG) and is presented as mean value +/- sem. n≥3. One-tailed student's t-test * p<0.05. (C) N2A cells were transfected with a plasmid co-expressing dCas9-KRAB and a sgRNA targeting either the TCF7L2 motif within each candidate CRE or a non-TCF7L2 motif containing sequence. A non-targeting sgRNA was used as a negative control. *Paupar* and *Pax6* expression levels were measured using RT-qPCR and results were normalised to *Gapdh*. Results are presented as mean values +/- sem, n≥3. One-tailed student's t-test * p<0.05, ** p<0.01, *** p<0.001. (D) Recruitment of MS2 tagged *Paupar* to a UAS *Pax6 Firefly* luciferase promoter reporter using a Gal4-MCP fusion protein up-regulates *Pax6* promoter activity. A *Renilla* expression vector was used as a transfection control and the total amount of DNA transfected in each case was made equal. Results are presented as mean values +/- sem, n = 4. One-tailed student's t-test * p<0.05, ** p<0.01.

ESCs whilst sequences downstream of P1 are in closer proximity to the *Paupar* promoter in neurons compared to ESCs. This asymmetry in local chromatin organisation may be important for the rewiring of short-range CRE-promoter interactions controlling *Paupar-Pax6* co-expression in neurons, and are consistent with ENCODE ChIP-seq data [24] showing that the chromatin surrounding the *Paupar* and *Pax6* promoters is marked by an increase in open (H3K4me1, H3K4me3, H3K27ac) chromatin marks in E12.5 mouse forebrain tissue compared to ESCs (Figs 5B and S3).

As transcriptional regulatory elements can function over large genomic distances, we next analysed 1 MB genomic sequence surrounding the *Paupar-Pax6* locus using binned NG Capture-C data to map meso-scale changes in chromatin architecture between cell types. This detected a large up to 250 kb chromosomal region located approximately 350 kb downstream of the *Pax6* gene that contains an increased frequency of statistically significant interactions with the *Paupar* and *Pax6* promoters in differentiated neurons compared to ESCs (Fig 5A). This region corresponds to an equivalent region in the human genome containing multiple predicted long-range CREs that loop onto the human Pax6 promoter in Promoter Capture Hi-C experiments [33]. Furthermore, this region contains an increased number of H3K4me1, H3K4me3 and H3K27ac ChIP-seq peaks [24] in mouse forebrain tissue compared to ESCs (Figs 5B and S3). As these histone modifications are known to mark active regulatory regions, we predict that this distal domain may contain additional clusters of uncharacterised long-range CREs involved in *Paupar-Pax6* expression control in neuronal cells. Taken together, these data define both local and distal chromatin changes associated with *Paupar-Pax6* expression in neurons.

## Discussion

LncRNAs involved in brain development are frequently co-expressed with their adjacent protein coding genes. These lncRNA-mRNA pairs often function in the same biological processes and some CNS expressed lncRNAs modulate both the expression and transcriptional activity of their neighbouring protein coding genes. A greater understanding of the complex regulatory relationship controlling the expression and function of lncRNA-mRNA pairs in the brain is needed to further define their role in neuronal development and function.

In this study we used high resolution NG Capture-C to comprehensively define chromatin interactions important for *Paupar-Pax6* expression control. Our work revealed an intricate network of short-range *cis*-regulatory interactions with the *Paupar* and *Pax6* P0 and P1 promoters, including interactions with the previously characterised *Pax6* ectodermal, neuroretina, retinal progenitor and photoreceptor enhancers [8–10], as well as many candidate new CREs. The results classified a subset of shared short-range chromatin interactions with both the *Paupar* and *Pax6* promoters that are likely to be involved in regulating *Paupar-Pax6* co-expression

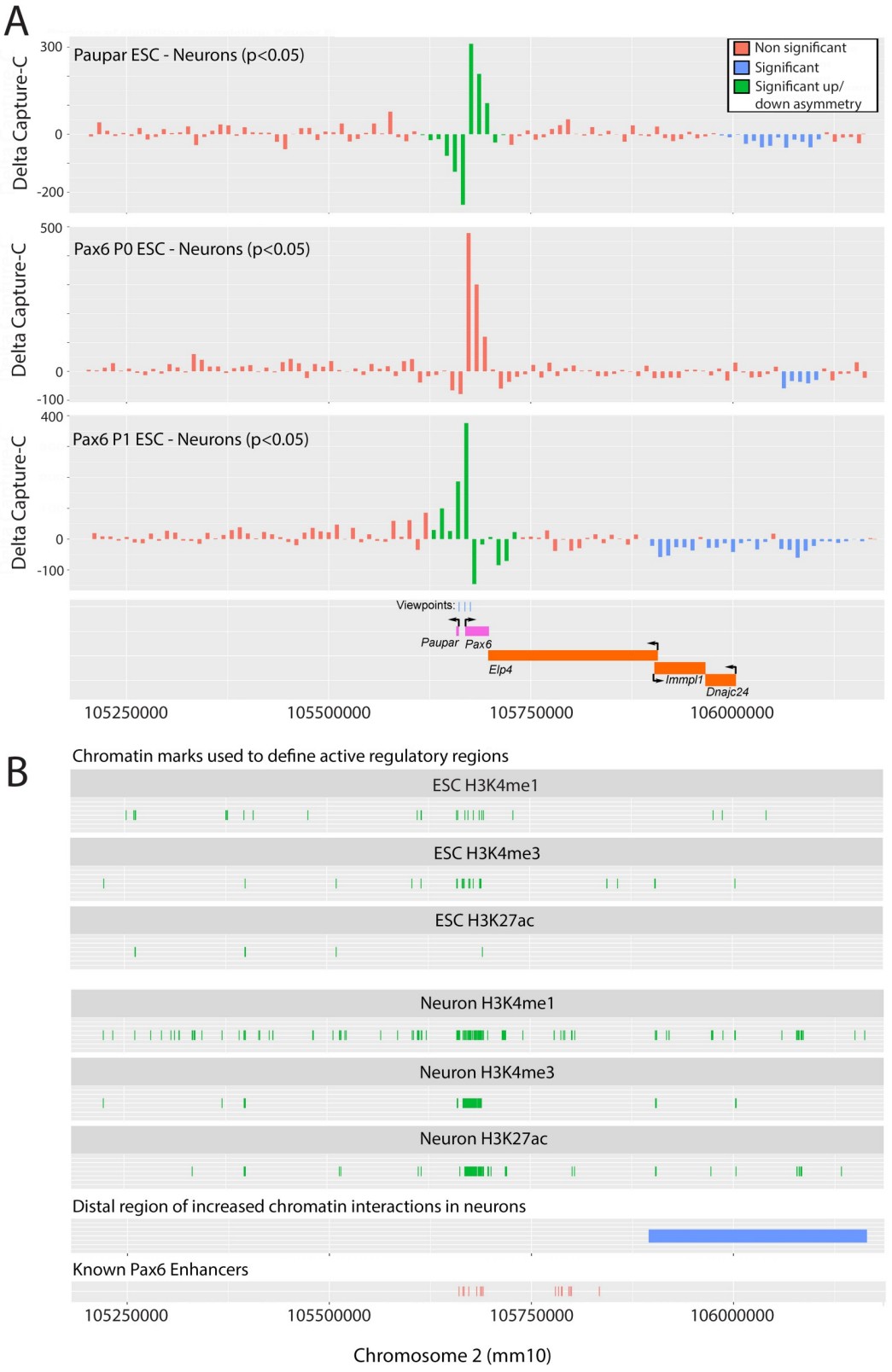

**Fig 5. Local and distal chromatin changes associated with increased *Paupar-Pax6* expression in neuronal cells.** (A) A comparative analysis of NG Capture-C data to map changes in chromatin conformation between cell types. Differences in mean normalized capture counts for interactions with the *Paupar* (top), *Pax6* P0 (middle) and *Pax6* P1 (bottom) viewpoints

between ESC and neurons are plotted on the y-axis. X-axis shows position on chromosome 2 (GRCm38/mm10). Data are binned to 10 kb across approximately ±500 kb genomic sequence around each viewpoint. Permutation testing was performed to determine statistical significance as described in Materials and Methods. (B) ENCODE Project ChIP-seq data mapping the location of H3K4me1, H3K4me3 and H3K27ac peaks in ESCs and mouse forebrain tissue across approximately 1MB genomic sequence surrounding the *Paupar-Pax6* locus [24]. Individual peaks of less than 1 kb are shown at 1 kb long for visibility reasons.

in the brain. We detected significant asymmetry in local chromatin architecture surrounding the *Paupar* and *Pax6* P1 promoters in differentiated neurons and ESCs and hypothesize that these changes are important for rewiring CRE-promoter communication and cell type specific expression of the locus.

We curated a subset of NG Capture-C fragments, based on increased local statistical significance, that may be central mediators of short-range CRE-promoter interactions. TCF7L2 binds a subgroup of these fragments and integrates signals from both the Wnt and Bmp signalling pathways to control *Paupar* and *Pax6* co-expression. Interplay between the Wnt and Bmp pathways is critical for proper development of the nervous system and has been shown to regulate postnatal NSC self-renewal and neurogenesis in the *Paupar-Pax6* expressing subventricular zone NSC niche [20,34,35]. Furthermore, the Wnt-Bmp signalling axis also promotes growth suppression and differentiation in neuroblastoma [29]. TCF7L2 is a key Wnt effector in the brain and is required for the production of *Pax6* expressing neural progenitor cells in the neocortex [36]. We expect that TCF7L2 acts as an important regulator of *Paupar-Pax6* chromatin organisation and CRE-promoter communication. Accordingly, TCF7L2 silencing leads to genome-wide changes in chromatin architecture and enhancer-promoter interactions in pancreatic and colon cancer cells, whilst TCF-bound Wnt responsive enhancers regulate chromatin looping and activation of the *Myc* gene in colorectal cancer [37–39]. Furthermore, ENCODE data shows that TCF7L2 associates with more than 40% of active enhancers in the genome in different cell lines suggesting that TCF7L2 is a critical regulator of cell-type specific CRE function [40]. ChIP-seq analysis could thus build on our results and map TCF7L2 chromatin occupancy across the *Paupar-Pax6* regulatory region at high resolution in *Paupar-Pax6* low- and high-expressing cells.

Our knockdown and pathway manipulation experiments indicate that the overall output of TCF7L2 activity is to up-regulate *Paupar-Pax6* expression. However, CRISPRi mediated annotation of motif containing sequences showed that TCF7L2 bound CREs can act as either enhancers or repressors of *Paupar* and *Pax6* co-expression. Although dCas9-KRAB recruitment induces the formation of a local closed chromatin state that will affect the accessibility of other nearby transcription factors binding sites, we propose a model in which different TCF7L2 motifs activate or repress *Paupar-Pax6* in a context dependent manner. This leads to precise control of *Paupar-Pax6* co-expression and is likely mediated through differential interactions between TCF7L2 and corepressor and coactivator proteins as described in [41].

Our work also discovered a large chromatin domain downstream of the *Pax6* gene that displays clusters of increased long-range chromatin interactions with the *Paupar-Pax6* locus in E14.5 mouse neurons compared to ESCs. This region is further downstream from the previously defined *Pax6* distal DRR and the aniridia-associated breakpoints within the last intron of the downstream *ELP4* gene that are predicted to influence *PAX6* expression in affected individuals [12, 21]. However, it maps to an equivalent region in the human genome that contains multiple long-range enhancer-promoter looping interactions with the *PAX6* promoter, is characterised by an increase in enhancer-like chromatin modifications and is located with within the same self-interacting TAD as the *Paupar* and *Pax6* promoters in neurons [33,42,43]. Our results are thus consistent with a model in which developmentally regulated

changes in distal chromatin architecture also play a role in CRE-promoter rewiring and the activation of *Paupar* and *Pax6* expression in the neuronal lineage.

This study further describes the function of the *Paupar* transcript and its DNA locus in *Pax6* expression control. We previously showed that shRNA mediated depletion of *Paupar* transcript induced a dose-dependent increase in *Pax6* expression [19]. However, this may be caused by transcriptional derepression of PAX6 mediated negative auto-regulation, as described in [32], as *Paupar-Pax6* are co-expressed in neuronal cells and *Paupar* mostly activates the expression of its direct transcriptional targets genome-wide [19]. Indeed, we demonstrate here that *Paupar* transcript has intrinsic transcriptional activator function and propose a model in which *Paupar* activates *Pax6* expression in a transcript-dependent manner whilst PAX6 can bind and negatively regulate its own promoter. NG Capture-C also revealed reproducible *cis*-regulatory interactions between DNA sequences within the *Paupar* locus and the *Pax6* P0 promoter in neuronal cell types, consistent with phase III ENCODE data showing that *Paupar* overlaps five candidate CREs with an enhancer-like chromatin signature [44]. We identify a TCF7L2 bound transcriptional silencer, CRE5, within the *Paupar* DNA locus and show that recruitment of dCas9-KRAB to the TCF7L2 motif increased both *Paupar* and *Pax6* expression. As this motif lies approximately 2 kb downstream of the *Paupar* TSS dCas9-KRAB recruitment would not block *Paupar* transcription [45]. This indicates that CRE5 represses both *Paupar* and *Pax6* expression independently of the *Paupar* transcript produced. Similarly, several studies have reported distinct roles for lncRNA transcripts and transcriptional regulatory elements within their DNA loci in gene expression control. The *Haunt* DNA locus contains several transcriptional enhancer elements that loop onto the *HoxA* gene to increase its expression whereas the *Haunt* transcript binds upstream of *HoxA* to induce a repressive chromatin state and block *HoxA* expression [46]. *Tug1* DNA contains a CRE that represses multiple neighbouring downstream genes whilst the *Tug1* lncRNA acts in *trans* to regulate different genes [47].

The identification of shared *Paupar* and *Pax6* CREs also raises the possibility that the *Paupar* promoter may be able to control *Pax6* expression through CRE competition as described for several other lncRNAs. The promoters of the *Pvt1* lncRNA and neighbouring *Myc* oncogene compete for interactions with four shared enhancers. Silencing the *Pvt1* promoter using CRISPRi increased enhancer contacts with the Myc promoter and up-regulated Myc expression independent of the Pvt1 transcript [48]. Similarly, the *Handsdown* locus interacts with several enhancers for the adjacent *Hand2* gene and regulates their usage during cardiac differentiation [49]. Our study provides significant new insights into the chromatin interactions, transcription factors and signalling pathways controlling *Paupar-Pax6* co-expression in the neuronal lineage and has general importance for understanding the wider role of lncRNA-mRNA transcription units in neuronal commitment, differentiation and function.

## Materials and methods

### Ethics statement

Primary cortical neurons and cortical neural stem cells were prepared from CD1 mouse embryos (E14.5) in accordance with UK Home Office Guidelines as stated in the Animals (Scientific Procedures) Act 1986 using Schedule 1 procedures approved by the University of Bath Animal Welfare and Ethical Review Body (NL1911-3).

### Plasmids

Individual sgRNAs were cloned into pX-dCas9-mod-KRAB to generate plasmids for CRISPRi as described in [50]. Oligonucleotides used to clone sgRNAs targeting the TCF7L2 motifs and

control elements are shown in S8 Table. The plasmid pCI-neo beta catenin S33Y was a gift from Bert Vogelstein (Addgene plasmid # 16519; http://n2t.net/addgene:16519; RRID: Addgene_16519). A 2.5 kb genomic region upstream of *Pax6* coding sequence was PCR amplified from N2A genomic DNA and subcloned into pGL4.10 (Promega) as a *Nhe*I-*Bgl*II fragment to generate the pGL4-Pax6P0 reporter. To generate the pUAS-Pax6P0Pro luciferase reporter a *Kpn*I-*Nhe*I fragment containing 9 repeats of the GAL4 UAS sequence was subcloned from pGL4.35 (Promega) into pGL4-Pax6Pro. 12 repeats of the MS2 binding site were PCR cloned as a *Not*I-*Spe*I fragment from pSL-MS2-12X (Addgene Plasmid 27119, [51]) into mEntry (GU931384). Full length *Paupar* was PCR amplified from N2A cell cDNA and inserted downstream as a *Spe*I-*EcoR*I fragment to generate mEntry-12xMS2-*Paupar*. The MCP was PCR cloned as a *Spe*I-*Asc*I fragment from pG14-MS2-GFP (Addgene Plasmid 27117, [51]) into mEntry. The Gal4 DNA binding domain (amino acids 1–147) was PCR cloned as a *Not*I-*Spe*I fragment from pSG424 [52] upstream and in frame of MCP to generate mEntry-Gal4DBD-MCP. The 12xMS2-*Paupar* and Gal4DBD-MCP sequences were then recombined into pcDNA3.2/V5-DEST using Gateway Technology (Invitrogen) for expression in mammalian cells.

## Cell culture

Cortices were dissected from embryonic brain and mechanically dissociated in PBS supplemented with 33 mM glucose, using a fire-polished glass Pasteur pipette.

## Primary neurons

For preparation of differentiated cortical neurons [53], cells were plated into Nunc 90 mm petri dishes, previously coated with 20 µg/ml poly-D-lysine (Sigma), at a seeding density of 500 x$10^5$ cells/ml. Neurons were cultured in Neurobasal medium (phenol red free) supplemented with 2 mM glutamine, 100 µg/ml penicillin, 60 µg/ml streptomycin and B27 (all from Gibco), and incubated at 37˚C, in high humidity with 5% $CO_2$. Under these growth conditions at 7 days *in vitro* (DIV) cells were non-dividing, had a well-developed neuritic network and were 99% β-tubulin III positive and <1% GFAP positive.

## Primary neural stem cells

For preparation of cortical neural stem cells [54], cells were plated into Nunc 90 mm petri dishes, previously coated with Cell Start (Gibco), at a seeding density of 500 x$10^5$ cells/ml. Neural stem cells were cultured in StemPro NSC SFM composed of: Knockout D-MEM / F12; Glutamax (2mM); bFGF (20ng/ml); EGF (20ng/ml); StemPro Neural Supplement (2%); all from (Gibco). Under these growth conditions at 7 DIV cells were proliferative and were Nestin and Ki67 positive.

## Cell lines

N2A cells were grown in DMEM supplemented with 10% fetal bovine serum (FBS). For the mouse ESCs experiments E14Tg2A cells were maintained in GMEM supplemented with 10% FBS, 1xMEM nonessential amino acids, 2 mM glutamax, 1 mM sodium pyruvate, 100 mM 2-mercaptoethanol, and 100 units/ml LIF on gelatinised tissue culture flasks.

## Transfections and treatments

Approximately 3 x $10^5$ N2A cells were seeded per well in a 6-well plate for both plasmid DNA and esiRNA transfections. The following day, cells were transiently transfected using

Lipofectamine 2000 (Invitrogen) following the manufacturer's instructions. For knockdown experiments, cells were transfected with 1.5 µg MISSION esiRNAs (Sigma-Aldrich) targeting either *Tcf7L2*, *Mash1* (*Ascl1*) *Klf16*, *Arid3a*, *Sox6* or *Renilla* Luciferase (EHURLUC) control and harvested 3 days later. 2µg pCI-neo beta catenin S33Y or empty vector were used in B-catenin S33Y overexpression experiments and cells were harvested 48 hrs post transfection. CRISPRi experiments were carried out as described in [50].

For Wnt, Rspo2 and Bmp treatment, approximately $3 \times 10^5$ N2A cells were seeded per well in a 6-well plate in growth medium containing either 100 ng/µl WNT3A or RSPO2 (both R&D Biosystems), or in low serum medium (DMEM supplemented with 5% FBS) containing 5ng/ml BMP4 (Thermo Fisher, PHC9534). 0.02% BSA in PBS was used as a vehicle control. Cells were harvested for RNA extraction 72 hours later. Sequences of primers used for expression analysis are shown in S8 Table.

Approximately $5 \times 10^4$ N2A cells were seeded per well in a 12-well plate for luciferase assays. The next day cells were transfected with 100ng reporter construct and 100ng GAL4-MCP and 400 ng MS2-*Paupar* expression vectors using FuGENE 6 (Promega) according to the manufacturer's instructions. The pRL-CMV plasmid (Promega) was co-transfected into each well to normalize for transfection efficiency. The total amount of DNA was made up to 1 µg for each transfection by the addition of empty expression vector. 2 days after transfection lysates were prepared and assayed for *Firefly* and *Renilla* luciferase activity.

## NG Capture-C

NG Capture-C libraries were prepared as described previously [3]. Briefly, approximately $2 \times 10^7$ cells per sample were fixed with 2% formaldehyde for 10 min at RT, quenched by the addition of glycine and washed with PBS. Cell lysis was performed for 20 min on ice (10 mM Tris-HCl, pH8, 10 mM NaCl, 0.2% IGEPAL), lysed cells were then homogenized on ice and enzymatically digested overnight at 37°C with DpnII (New England Biolabs). The digested DNA was diluted and ligated with T4 DNA ligase overnight at 16°C. The following day, ligation reactions were de-crosslinked by Proteinase K (Thermo Scientific) addition and overnight incubation at 65°C. DNA extraction was performed by Phenol/Chloroform/Isoamyl Alcohol and chloroform/isoamyl alcohol extraction followed by ethanol precipitation. DpnII digestion efficiency was confirmed by gel electrophoresis and quantified by real-time PCR–only 3C libraries with over 70% efficiency were used for the subsequent steps. Samples were sonicated to an average size of 200 bp using a Bioruptor Pico (Diagenode) and NEBNext Multiplex reagents and sequencing adapters were used to prepare sequencing libraries following the Illumina NEBNext DNA library prep kit instructions (New England Biolabs). Two rounds of capture using a pool of biotinylated oligos (IDT, see S8 Table for sequences) were performed on 1ug of each of the indexed libraries using the Nimblegen SeqCap EZ hybridization system. Library size was determined using the Tapestation D1000 kit and the DNA concentrations were measured on a Qubit 2.0 Fluorometer.

## Computational analysis of NG Capture-C data

Multiplexed NG Capture-C libraries were prepared from ESCs, NSCs, neurons and N2A cells (two biological replicates each) and 150 bp paired end sequencing was performed on the Illumina HiSeq 4000 (Novogene) to a total depth of approximately 500 million reads. The resulting fastq files from each of the eight replicates were combined using a Perl script. Raw reads were trimmed using trim_galore version 0.4.4 with parameter –paired. The trimmed paired-end reads were then combined using flash version 1.2.11 with the parameters —interleaved-output –max-overlap = 200. The resulting fastq files of combined and uncombined reads were

next merged into a single fastq file using the command cat. The fragments in the resulting fastq file were *in silico* digested into DpnII restriction enzyme digestion fragments using dpnII2E.pl. The resulting dpnIIE fragments were aligned to the mm10 genome using bowtie version 1.1.2 with parameters -p 1 -m 2 —best—strata —chunkmb 256 –sam. A set of DpnII fragments for the full mouse genome was produced from mm10.fa using gpngenome.pl. CCAnalyser3.pl was then run to compare a text file of the *Paupar*, *Pax6* P0, *Pax6* P1 and *Sox2* viewpoint coordinates with the *in silico* digested reads and genome to produce counts of the observed interactions with each viewpoint in each replicate. The output from CCAnalyser3.pl was analysed using the BioConductor r3Cseq package to determine statistical significance (p- and q-values) for the observed interactions between the viewpoints and each DpnII digestion fragment in each replicate. For each viewpoint, the resulting tables were combined into a table showing replicated significant fragments (significant in both replicates of at least one cell type) and these tables were used as the basis for subsequent analyses. Motif discovery was performed using the BiFa web tool at the Warwick Systems Biology Centre website (http://wsbc.warwick. ac.uk/wsbcToolsWebpage). 4Cin was used to generate three-dimensional models of *Paupar-Pax6* local chromatin architecture [26].

## A statistical method for detecting meso-scale changes in chromatin conformation from NG Capture-C data (DeltaCaptureC)

We developed a new statistical method to detect changes in chromatin conformation based on significant clustering of neighbouring fragments from 3C-based data. This is available as a Bioconductor Software Package (10.18129/B9.bioc.deltaCaptureC). By binning NG Capture-C data and using permutation testing, this package can test whether there are statistically significant changes in the interaction counts between the data from two cell types or two treatments. To do this, read counts from the two biological replicates for each cell type were first combined. The counts for the four samples were normalised using DESeq2 function estimateSizeFactorsForMatrix() [55] and the mean normalised count for both replicates in each cell type was determined. The difference between the two mean normalized counts was then calculated. This data was trimmed to a region of interest, 500kb up- and down-stream of the midpoint of viewpoint, binned to a fixed bin size of 1kb and then re-binned to 10kb (S2 Fig). This identified a large distal region of increased chromatin interactions in neurons (Figs 5 and S2). We observed that this a contiguous region of constant sign (negative) with a combined total absolute value of 308.8. The null hypothesis is that this sum arises by chance. We tested this hypothesis to detect statistical significance for continuous regions of constant sign in the following manner: we first excluded the region 50kb up- and downstream of the viewpoint and performed random permutation of the (non-viewpoint) 1kb bins. After each such permutation, data was re-binned to 10kb and each region was examined for constant sign. To do this, we computed its total absolute value and recorded the largest of these totals. If, after performing 1000 such random permutations, we observe fewer than 50 cases where the largest sum is 308.8 or greater, we have discovered a p-value for this region of less than 0.05. In this way, we can exploit co-localisation of differences with like sign to detect meso-scale chromatin remodelling from 3C-based data (Figs 5A and S2).

We then considered the region near the viewpoint. In this case it is important to note that raw NG Capture-C counts in this region strongly correlate with distance from the viewpoint and we are thus unable to perform arbitrary permutation to test for statistical significance. However, performing permutations which do not change this distance allowed us to test the null hypothesis that chromatin remodelling flanking the viewpoint was symmetric. To this end, we computed the difference between the sum in the region upstream of the viewpoint

and downstream of it using the actual data. We then computed this difference after multiple symmetric permutations. Since there are 50 1kb bins in this region upstream and downstream of the viewpoint, there are $2^{50}$ permutations of this form giving us enough for permutation testing. In this way, we detected asymmetry in chromatin architecture in the neighbourhood of the *Paupar* and *Pax6* promoter viewpoints in each cell type (Fig 5A).

## RT-qPCR

RNA extraction was carried out using the GeneJET RNA Purification Kit (ThermoFisher) according to the manufacturer's instructions with the addition of an on-column DNase digestion step using the RNase-free DNase Set (QIAGEN). Reverse transcription was performed using the QuantiTect Reverse Transcription Kit (Qiagen). 1 µg total RNA was used in each reaction. Quantitative PCR was carried out on a Step One Plus Real-Time PCR System using Fast SYBR Green Master Mix (Applied Biosystems).

## Chromatin immunoprecipitation

ChIP experiments were performed as previously described using approximately 1 x $10^7$ N2A cells per assay [19]. Cross-linked chromatin was immunoprecipitated with either 5 µg anti-TCF4/TCF7L2 (Clone 6H5-3, #05–511, Millipore) or normal mouse control IgG (#12–371, Millipore) antibodies. qPCR primers used to amplify TCF7L2 motif containing sequences (CRE1-7) are shown in S8 Table.

## Supporting information

**S1 Fig. NG Capture-C identified ESC-specific chromatin looping interactions between the *Sox2* super-enhancer spanning the *Peril* locus and the *Sox2* promoter.** NG Capture-C profiles displaying the *Sox2* promoter interaction count per DpnII restriction enzyme fragment in the indicated cell types. The red vertical line indicates the location of the *Sox2* promoter viewpoint. Significant interactions were determined using r3C-seq.
(TIF)

**S2 Fig. Increasing bin size facilitates the detection of chromatin changes between cell types.** Differences in mean normalized NG Capture-C counts between neurons and ESCs for interactions with the *Paupar* viewpoint are plotted on the y-axis. X-axis shows position on chromosome 2 (GRCm38/mm10). Sequence data was permuted to assess specificity and statistical significance was calculated as described in Materials and Methods. (A) Difference in mean normalised interactions between ESCs and neurons binned to 1 kb. Negative values shown in red indicate increased interactions in neurons. Positive values in turquoise illustrate increased interactions in ESCs. (B) The same data binned to 10 kb. Note the emergence of the large red region approximately 350 kb downstream of *Pax6* and the asymmetric pattern near the viewpoint. (C) The 1 kb bins from (A) permuted. Bins further than 50 kb from the viewpoint are permuted at random. Bins closer than 50 kb are only permuted keeping their distance from the viewpoint. (D) The previous panel re-binned to 10kb. Notice that no large contiguous regions of constant sign appear, nor does the asymmetric pattern seen near the viewpoint in (B).
(TIF)

**S3 Fig. An increased frequency of chromatin interactions with the *Paupar* and *Pax6* promoters in neurons correlates with elevated levels of open compared to closed histone modifications.** ENCODE Project ChIP-seq data mapping the location of open (H3K4me1, H3K4me3 and H3K27ac) and closed (H3K27me3 and H3K9me3) ChIP-seq peaks in ESCs (A)

and E12.5 mouse forebrain tissue (B) across approximately 1MB genomic sequence surrounding the *Paupar-Pax6* locus [24]. Individual peaks of less than 1000 bp are shown at 1000 bp long for visibility reasons. Aggregated data represent summed and binned scores from the individual tracks.
(TIF)

**S1 Table. Determination of ligation and capture frequencies for NG Capture-C experiment.**
(XLSX)

**S2 Table. Number of unique interactions with each promoter viewpoint.**
(XLSX)

**S3 Table. Genome coordinates of significant replicated fragments for each viewpoint (Mouse GRCm38/mm10).**
(XLSX)

**S4 Table. Permutation testing reveals significant overlap between NG Capture-C chromatin interactions with the *Pax6* promoters and H3K4me1 mouse forebrain ChIP-seq peaks.**
(XLSX)

**S5 Table. Interactions between the indicated viewpoints and reporter fragments that overlap the *Paupar* genomic locus.**
(XLSX)

**S6 Table. Number of replicated interactions in different cell types.**
(XLSX)

**S7 Table. Bioinformatics identification of transcription factor motifs.**
(XLSX)

**S8 Table. Oligonucleotides.**
(XLSX)

**S9 Table. Numerical data used to generate individual figures.**
(XLSX)

## Acknowledgments

We thank Dr Karim Malik (University of Bristol) for providing recombinant Wnt3a, Rspo2 and Bmp4, and University of Bath Final Year Project Students Tazmin Martin and Hera Wong for help cloning CRISPRi constructs.

## Author Contributions

**Conceptualization:** Michael Shapiro, Keith W. Vance.

**Data curation:** Ioanna Pavlaki, Michael Shapiro, Keith W. Vance.

**Formal analysis:** Ioanna Pavlaki, Michael Shapiro, Keith W. Vance.

**Funding acquisition:** Keith W. Vance.

**Investigation:** Ioanna Pavlaki, Michael Shapiro, Giuseppina Pisignano, Stephanie M. E. Jones, Robert J. Williams, Keith W. Vance.

**Methodology:** Ioanna Pavlaki, Michael Shapiro, Jelena Telenius, Silvia Muñoz-Descalzo, Robert J. Williams, Jim R. Hughes, Keith W. Vance.

**Project administration:** Michael Shapiro, Keith W. Vance.

**Resources:** Michael Shapiro, Jelena Telenius, Silvia Muñoz-Descalzo, Robert J. Williams, Jim R. Hughes, Keith W. Vance.

**Software:** Michael Shapiro, Jelena Telenius, Jim R. Hughes.

**Supervision:** Robert J. Williams, Keith W. Vance.

**Validation:** Ioanna Pavlaki, Michael Shapiro, Keith W. Vance.

**Visualization:** Ioanna Pavlaki, Michael Shapiro, Keith W. Vance.

**Writing – original draft:** Keith W. Vance.

**Writing – review & editing:** Ioanna Pavlaki, Michael Shapiro, Giuseppina Pisignano, Silvia Muñoz-Descalzo, Robert J. Williams, Keith W. Vance.

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
