## [Decision Letter · Decision Letter 0]

2 May 2022

Dear Dr Vance,

We are pleased to inform you that your manuscript entitled "Chromatin interaction maps identify Wnt responsive cis-regulatory elements coordinating Paupar-Pax6 expression in neuronal cells" has been editorially accepted for publication in PLOS Genetics. Congratulations! Find the reviewer's comments below and respond to the minor comment raised by Reviewer 2 regarding limitations of the study.

Yours sincerely,

Ivan D'Orso, PhD

Guest Editor

PLOS Genetics

John Greally

Section Editor: Epigenetics

PLOS Genetics

Comments from the reviewers:

Reviewer #1: The authors have sufficiently addressed my critiques in this revision.

Reviewer #2: In the revised version of their manuscript entitled “Chromatin interaction maps identify Wnt responsive cis-regulatory elements coordinating Paupar-Pax6 expression in neuronal cells”, Vance and colleagues have addressed most of the criticism that was raised. While I still consider the lack of a genome-wide assessment of TCF7L2 a detriment to the whole story, I appreciate the authors’ effort to increase the robustness of the CRISPRi approach. I also appreciate that they now provide a justification for diving into TCF7L2 as a candidate TF. They further include a reporter assay with MS2 fused to Paupar that adds more depth to their overall findings.

**Have all data underlying the figures and results presented in the manuscript been provided?**

Reviewer #1: Yes

Reviewer #2: Yes

PLOS authors have the option to publish the peer review history of their article (what does this mean?). If published, this will include your full peer review and any attached files.

Reviewer #1: No

Reviewer #2: No

**Data Deposition**

http://datadryad.org/submit?journalID=pgenetics&manu=PGENETICS-D-22-00391

**Press Queries**

---

## [Editor Report · Acceptance letter]

26 May 2022

PGENETICS-D-22-00391 

Chromatin interaction maps identify Wnt responsive cis-regulatory elements coordinating Paupar-Pax6 expression in neuronal cells 

Dear Dr Vance, 

We are pleased to inform you that your manuscript entitled "Chromatin interaction maps identify Wnt responsive cis-regulatory elements coordinating Paupar-Pax6 expression in neuronal cells" has been formally accepted for publication in PLOS Genetics! Your manuscript is now with our production department and you will be notified of the publication date in due course.

With kind regards,

Anita Estes

PLOS Genetics

On behalf of:
